# Oral Administration as a Potential Alternative for the Delivery of Small Extracellular Vesicles

**DOI:** 10.3390/pharmaceutics15030716

**Published:** 2023-02-21

**Authors:** Darío Donoso-Meneses, Aliosha I. Figueroa-Valdés, Maroun Khoury, Francisca Alcayaga-Miranda

**Affiliations:** 1Laboratory of Nano-Regenerative Medicine, Centro de Investigación e Innovación Biomédica (CiiB), Faculty of Medicine, Universidad de los Andes, Santiago 7620086, Chile; 2IMPACT, Center of Interventional Medicine for Precision and Advanced Cellular Therapy, Santiago 7620086, Chile; 3Consorcio Regenero, Chilean Consortium for Regenerative Medicine, Santiago 7550101, Chile; 4School of Medicine, Faculty of Medicine, Universidad de los Andes, Santiago 7620086, Chile; 5Cells for Cells, Santiago 7620157, Chile

**Keywords:** small extracellular vesicles, milk-derived vesicles, food-derived vesicles, exosomes, oral delivery, oral administration, oral drug delivery, biodistribution

## Abstract

Small extracellular vesicles (sEVs) have burst into biomedicine as a natural therapeutic alternative for different diseases. Considered nanocarriers of biological origin, various studies have demonstrated the feasibility of their systemic administration, even with repeated doses. However, despite being the preferred route of physicians and patients, little is known about the clinical use of sEVs in oral administration. Different reports show that sEVs can resist the degradative conditions of the gastrointestinal tract after oral administration, accumulating regionally in the intestine, where they are absorbed for systemic biodistribution. Notably, observations demonstrate the efficacy of using sEVs as a nanocarrier system for a therapeutic payload to obtain a desired biological (therapeutic) effect. From another perspective, the information to date indicates that food-derived vesicles (FDVs) could be considered future nutraceutical agents since they contain or even overexpress different nutritional compounds of the foods from which they are derived, with potential effects on human health. In this review, we present and critically analyze the current information on the pharmacokinetics and safety profile of sEVs when administered orally. We also address the molecular and cellular mechanisms that promote intestinal absorption and that command the therapeutic effects that have been observed. Finally, we analyze the potential nutraceutical impact that FDVs would have on human health and how their oral use could be an emerging strategy to balance nutrition in people.

## 1. Introduction

The enteral route, including oral administration of drugs is the preferred delivery method to treat systemic diseases or local gastrointestinal (GI) pathologies due to its minimal invasiveness (pain-free), relatively low cost, and ability to self-administer [1]. However, these advantages are challenged by acidic conditions in the stomach and degrading conditions in the intestine, which affect the stability, absorption, and bioavailability of various therapeutic molecules, limiting the diversity of therapeutic compounds that can be prescribed orally [2]. Indeed, macromolecules such as proteins, peptides, or nucleic acids as free agents show only slight absorption when administered orally, as they are degraded by GI enzymes, have low stability at acidic pH and limited permeation through biological barriers [3]. Likewise, several hydrophilic and lipophilic drugs also have limitations for their oral intake since their absorption is greatly conditioned by their molecular weight, logP value, and gastrointestinal permeability, often requiring nanocarriers to generate a biological effect [4].

Nanocarriers are a colloidal transport system for drugs with a nanometric particle size (typically < 500 nm) [5]. In their oral administration, they allow the safe transport of active therapeutic molecules, improving pharmacokinetics, biodistribution, and stability, reducing toxicities, and offering a controlled release and delivery of drugs to specific sites [5]. Small extracellular vesicles (sEVs) are one of the most studied nanocarriers due to their biological origin that endows them with different and natural attributes that favor their biomedical use [6]. Defined as cell-derived nanostructures enclosed in a lipid membrane, they transport and protect an active molecular cargo composed of nucleic acids, proteins, and lipids [7,8,9,10]. In the physiologic and pathologic processes, they play a role in the regulation of intercellular communication [11,12]. As nanocarriers, they have advantageous attributes related to their proven rapid internalization, low immunogenicity even at repeated doses, high stability in physiological conditions, and the capability of modifying their internal and superficial components to generate a controlled and specific release of endogenous or loaded therapeutic molecules [13]. Compared with synthetic nanocarriers, sEVs exhibit substantial benefits in targeting, safety, and pharmacokinetics, being considered the next-generation drug delivery platform [14].

Since its introduction as a drug delivery nanosystem, research has focused on understanding the pharmacokinetics and biodistribution of intravenously (i.v.) administered sEVs [15]. The valuable information collected on this subject contrasts with the need for more knowledge about the safety, stability, pharmacokinetics, and biodistribution of orally administered sEVs, the preferred route of administration for doctors and patients. Although there are few quality studies, the data to date show that, remarkably, sEVs can withstand the harsh environment of the GI tract and reach the intestine, where they accumulate heavily. It is from the intestinal lumen where sEVs penetrate the epithelium, which is the innermost layer lining the entire GI tract and selectively regulates transport from the lumen to the underlying tissue compartment [1]. Although most of these studies are limited to a single source of sEVs, cow’s milk, they remarkably demonstrate the efficacy of using sEVs as a nanocarrier system for a therapeutic payload to elicit a desired biological effect.

In this review, we first discuss the current evidence on the challenges of the ambitious concept of using sEVs as a nanocarrier system for oral prescription, focusing on the stability, bioavailability, and uptake of sEVs by intestinal cells. Second, we present a comparative analysis summarizing the biodistribution and toxicity of orally administered sEVs in murine models. Then, we investigated through the available data molecules or motifs that could explain the endogenous capacity of certain sEVs to transit the GI pathway and cause a therapeutic response, addressing the possible cellular mechanisms by which this response would be mediated. Finally, we discuss the nutraceutical perspective of the consumption of bioactive molecules within sEVs derived from food sources and their potential impact on human health.

## 2. Challenges of Orally Administered sEVs

sEVs are non-replicative lipid-based vesicles classified by a hydrodynamic diameter inferior to 200 nanometers (nm) [16]. They are secreted by most known cells and can be found in every biological fluid (blood, urine, saliva, breast milk, among others) [13]. Besides, their biogenesis mechanism allows the transport and protection of bioactive molecules (nucleic acids, proteins, lipids, or metabolites) from a donor towards an acceptor cell, modifying their transcriptional profile, function, or phenotype [13]. Unique properties such as high relative stability, biocompatibility, permeability, low toxicity, and low immunogenicity determine its success as a novel cell-free therapeutic agent. Their versatile properties may be modified by different bioengineering allowing the insertion of targeting motifs for specific cellular lineages in their surface, a load of therapeutic drugs or macromolecules in their membrane or lumen, or even they can be modified to increase blood circulation time (Figure 1) [17].

Orally administered sEVs must survive the harsh degrading conditions of the digestive system as moisture, lubricants, mechanical forces, digestive enzymes, emulsifiers, pH neutralizers [2], commensal microbiota and their derivates [18] to successfully reach the intestine and deliver their therapeutic payload regionally or be absorbed intact for systemic distribution (Figure 2). The latter is the most challenging since once the sEVs penetrate the intestinal mucus layer, they must cross the intestinal epithelium to reach the lamina propria and cross the endothelium of blood vessels for systemic distribution [19]. When synthetic nanoparticles are orally ingested, most are degraded or eliminated, and a small fraction is effectively absorbed [20]. Nonetheless, it is currently unknown if the same proportion of orally administrated sEVs is degraded or eliminated since sEVs differs in their surface molecules, expressing receptors, peptides, saccharides, and lipids from a biological progenitor.

To date, it has been described that different types of nanoparticles can cross the intestinal epithelium using different mechanisms, such as paracellular transport [1]. Paracellular transport consists of the diffusion of particles between cells through tight junctions that form the intestinal epithelial barrier. However, due to limited physical dimensions between cells in physiological conditions, only particles ranging between 0.5 and 20 nm should be considered for this mechanism in a relevant proportion [21]. Conversely, a pro-inflammatory context in the intestine disrupts the epithelial barrier, allowing the passing of larger particles, as demonstrated by Tulkens et al. [22]. Thus, based on the reported results, the paracellular transport of an oral administration of sEVs (>200 nm) should only be considered in inflammatory diseases of the intestine or treatments that disrupt the epithelial barrier as side effects.

The second mechanism described by which nanoparticles cross the intestinal layers and reach systemic circulation is transcellular transport by epithelial cells (mainly enterocytes that constitute 90–95% of the cells in the GI tract) and M-cells (specialized phagocytic cells that represent 1% of the intestinal epithelium) [1]. Transcellular transport consists of the endocytosis of particles in the apical face, the intracellular transit, and posterior exocytosis in the basal face [20]. The main challenge for this route of absorption is avoiding the transport and fusion of the intracellular vesicles with lysosomes enriched in degradative enzymes. A unidirectional transport of bacterial-derived extracellular vesicles (bEVs) was demonstrated by epithelial cells in vitro, where a small proportion was “re-secreted” towards the basal face, partially supporting the proof of concept of orally ingested sEVs potential absorption and lysosome avoiding [23]. A better understanding of the molecular mechanism of uptake by these intestinal cells could contribute to an engineered increased interaction with sEVs. Macropinocytosis, caveolin, and clathrin-dependent endocytosis are the principal mechanisms described for particle uptake in enterocytes. In M-cells, the most studied are the uptake by phagocytosis and receptor-mediated endocytosis [20].

## 3. Biodistribution, Stability, and Safety of Oral Delivery of Native and Drug Loaded sEVs

Murine studies identifying the biodistribution pattern of orally administered sEVs are few and focus mainly on cow’s milk-derived sEVs, although we have found some studies using plant-derived exosomes-like particles. These studies show that sEVs/exosomes-like particles manage to withstand the hostile environment of the gastrointestinal tract, associated with their transit through acidic conditions in the stomach and degradative conditions in the gut, in various murine models [24,25,26]. Cow’s milk-derived sEVs cross the upper gastrointestinal tract and reach the intestine in relatively short times (1–6 h) [27,28]. The absorption of sEVs seems to occur in the gut through mechanisms that are not well understood, but that facilitates the entry of sEVs into the systemic circulation and their distribution in other organs, essentially localized in the abdominal cavity [27,28,29,30,31]. Unlike the “trapping” of sEVs in the organs of the mononuclear phagocytic system (liver, spleen, and lung) after systemic injection of sEVs [17], oral ingestion allows a considerable accumulation of sEVs in the intestine [27,28,30,32,33,34]. In the other organs of the body, the accumulation of sEVs is less but notably shows a homogeneous distribution among them. Figure 3 shows the biodistribution pattern in mice after oral and systemic administration. 

Interestingly, repeated oral administration of cow’s milk-derived sEVs on mice seems to not alter the biodistribution pattern observed after a single oral intake of sEVs [28]. Betker et al. [29] and Samuel et al. [28] also show that orally ingested sEVs can migrate and accumulate in xenograft tumors in vivo. Other sources of sEVs studied in similar investigations are those obtained from yeast [28], beer [28], grape [34], acerola [33], ginger [32], garlic [35], tea leaves [36], and mulberry bark [37]. Yeast-, grape-, acerola-, ginger-, garlic-, tea leaves- and mulberry bark-derived sEVs showed a biodistribution pattern like that previously described for cow’s milk-derived sEVs, but beer-derived sEVs in the mouse’s organs could not be detected [28]. Interestingly, orally ingested ginger-derived exosomes-like particles showed a differential bio-distribution after 12 h of gavage depending on the feed condition of mice: starved mice accumulated exosomes-like particles in the stomach and small intestine, whereas non-starved mice accumulated exosomes-like particles in the colon [32]. These conditions open a new variable to consider for the pharmacokinetic profile of the oral administration of sEVs, loaded or not with drugs. Whether or not other cellular sources (of prokaryotic or eukaryotic origins) of sEVs can cross the harsh microenvironment of the gastrointestinal tract and replicate the biodistribution pattern described so far is still unknown. Table 1 summarizes the key aspects of the studies performed to determine the biodistribution pattern of sEVs/exosomes-like particles administered orally, including the type of sEVs, the cellular origin of the sEVs, doses of sEVs administered, time of detection, tissue distribution, among other variables of relevance.

As mentioned, the intestine is the anatomic site where the absorption of sEVs seems to occur after oral gavage in mice, which results in their entry into the bloodstream. Transendocytosis through intestinal epithelial cells [29,38] or paracellular translocation through the epithelial barrier [22] are some proposed mechanisms for this phenomenon. Figure 4 shows the mechanisms of cellular absorption of orally administered sEVs that are expected to occur in the human intestinal epithelium. This figure was elaborated based on data obtained from preclinical studies. Recently, Wu et al. [25] determined through a series of well-established experiments that cow´s milk derived sEVs loaded with insulin exhibited efficient internalization by active multiple endocytic routes to the epithelia. Since sEVs derived from the milk (nutrient), the authors also studied the involvement of the nutrient-assimilation pathway. The data showed that the uptake of milk-derived sEVs its mediated by peptide transporter, amino acid transporters, glucose transporters, and the neonatal Fc receptor (FcRn) [25], as was first proposed by Betker et al. [29]. However, the uptake of sEVs was not affected by Niemann-Pick C1-like 1 (NPC1L1) protein, which mediates the absorption of cholesterol and phytosterols [25]. According to Sriwastva et al. [37] mulberry bark-derived exosome-like particles were predominantly taken up by gut epithelial cells, Paneth cells, and colon tissue. Furthermore, in the spleen and liver, these particles were predominantly present in F4/80+ macrophages. In this work, mice showed no adverse effects, no significant changes in body weight, skin rashes, or abnormal fecal discharge, and no abnormal effects regarding morphology of internal organs, microscopic structure of gut tissue, blood cholesterol, triglycerides, or liver enzyme alanine transaminase. Due to the complex composition and structure of sEVs, more receptors and transporters need to be investigated to elucidate the endocytic mechanisms that facilitate uptake in intestinal epithelial cells. As well as, to validate the data obtained in in vitro experimental settings in in vivo conditions.

Little information exists about other pharmacokinetic parameters of orally administered sEVs. Regarding its stability in circulation, Munagala et al. [31] observed that cow’s milk-derived sEVs remained in circulation for at least 24 h after oral administration in nude mice. Results that strongly contrast with the numerous studies demonstrating the rapid clearance rate of circulating exogenous exosomes after systemic injection (~2–30 min), mainly mediated by macrophages [17]. Why sEVs absorbed from the gastrointestinal tract have a longer circulating half-life than observed in systemically administered sEVs is a question that must be answered to understand the real clinical potential of oral administration of sEVs. It is worth mentioning that the same research group subsequently tested milk-derived sEVs for oral administration of the chemotherapeutic drug paclitaxel (PAC) in a lung tumor xenograft model, demonstrating that orally administered PAC-loaded sEVs significantly inhibited tumor growth compared to the same dose of PAC administered intraperitoneally. These PAC-loaded sEVs showed remarkably lower systemic and immunologic toxicities as compared to i.v. PAC [39]. Soo Kim et al. [40] loaded murine RAW 264.7 macrophages-derived sEVs with PAC, showing a more than 50-fold increase of cytotoxicity in drug resistant MDCK_MDR1_ (Pgp+) cells in vitro.

The studies that evaluated the safety profile of sEVs orally administered consistently showed that the parameters of body weight, plasma cytokine concentration, or tissue damage remain unchanged, suggesting they are well-tolerated and non-immunogenic. Although these studies were performed mainly in milk-derived sEVs, different concentrations, and even at repeated doses, toxicity data confirm preliminarily the potential clinical use of milk-derived sEVs. Table 2 details the main results that allow knowing the safety profile of oral administration of sEVs. Nonetheless, we highlight the fact that vesicles derived from food sources may not be the exclusive sEVs that possess the capacity to be absorbed after oral ingestion, emphasizing the lack of information from other sEVs sources with proven therapeutic properties such as mesenchymal stem cells [41].

## 4. sEVs Attributes for an Efficient Oral Administration

Very few studies have investigated the use of sEVs formulations for gastric drug delivery. According to Bardonnet et al. [42], nanoparticle size is essential for gastric retention because particles with a diameter < 7 mm are efficiently evacuated. Since sEVs possess a much smaller size range of 50–200 nm [13], in their native state (unmodified) is unlikely to exert any biological effect in the stomach due to weak gastric retention. However, modifying sEVs with mucoadhesion strategies using polymers or phospholipids in their surface membrane could give them time to trigger the desired biological changes [1]. Regarding intestinal drug delivery, using unmodified sEVs as nanocarriers has demonstrated promising results. Several studies have reported systemic absorption of drugs in the intestine from sEVs or a regional effect, as described in the previous section. However, a wide array of modifications has also been tested to improve sEVs stability in the GI tract, uptake by intestinal cells, and even delivery to cells independent (or far from) of the GI system. Table 3 presents a summary of these articles, where they were classified according to the source of sEVs, the attribute or modification studied, the use of sEVs as a drug nanocarrier, and the observed biological effect and the type of models utilized. As a note to mention, most articles on the oral administration of sEVs are based on the use of vesicles derived from edible compounds (fruits, vegetables, spices, milk, or its derivatives), as extensively reviewed by Ciéslik et al. [24].

Of all these sources of sEVs, notably bovine milk has reported increased stability in acidic media that emulate the conditions of the stomach lumen and structural conservation after boiling due to the presence of calcium in comparison to colorectal cancer-derived sEVs (LIM1215 cells) [28]. Besides, the addition of casein (a highly abundant protein in breast milk) has been shown to enhance the uptake of sEVs isolated from human cardiosphere-derived stromal/progenitor cells after oral ingestion [43]. The modification of sEVs with casein also presented an increased biological effect than unmodified sEVs in cardiac dysfunction [43]. This data indirectly supports the bovine milk-derived sEVs as nanocarriers for oral drug delivery since the abundant natural presence of casein in the milk should confer similar properties to those isolated vesicles. Another compound present in breast milk from various species is folic acid [47]. In a publication from Munagala et al. [31], the addition of folic acid to the surface of sEVs isolated from bovine milk and loaded with withaferin A decreased the tumor volume in a murine model of lung cancer. The modification with folic acid in the sEVs surface increased the therapeutic effect in this cancer model compared to the unmodified sEVs; however, it is not clarified if this response is attributed to enhanced stability in the GI tract or if a targeting to tumor cells after systemic circulation is reached [31].

Another approach for increasing the uptake of milk-derived sEVs was published by Warren et al. [3], where they modified the surface of the vesicles with Polyethylene glycol (PEG). Due to this modification, hydrophobic interactions with mucin (present in the lumen of the intestine) are decreased, thus enhancing the interaction, uptake by epithelial cells, and delivery of a loaded siRNA in vitro. Besides, adding PEG to the surface of the milk-derived sEVs increased the recovery after incubation in acidic conditions mimicking an infant (pH 4.5) or adult (pH 2.2) stomach acidity [3].

Another studied source of sEVs with therapeutic properties after oral delivery is grape juice. Ju et al. [34] showed that grape exosome-like nanoparticles (GELNs) isolated from grape juice possess bioactivity in intestinal stem cells, protecting from colitis in an in vivo-induced model and facilitating organoid formation in vitro. They assembled liposome-like nanoparticles with lipids from these GELNs and showed their role in in vivo targeting of intestinal stem cells through oral gavage [34]. As mentioned previously, it is currently unknown how these modifications mentioned above could modify the bioavailability of sEVs isolated from other sources different from foods or their derivatives. 

## 5. Cellular and Molecular Mediators for sEVs Uptake after Oral Administration 

Despite the incipient understanding of the cellular/molecular mechanism that regulates the biological effect of sEVs through oral administration, in the literature, it is possible to find several articles reporting therapeutic efficacy when sEVs are administered orally in models of inflammatory diseases and cancer. 

In an induced cutaneous delayed-type hypersensitivity (DTH) model, Nazimek et al. [48] and Wasik et al. [49] demonstrated that T cells and B1a cells secrete a subpopulation of immunosuppressive sEVs that contain the inhibitory miRNA-150, which prevent inflammation and DTH after systemic administration in mice [48,49]. In the second of these publications [49] intravenous, intraperitoneal, intradermal, and oral administration of equivalents doses of the immunosuppressive sEVs were tested head-to-head to evaluate the anti-inflammatory response. Unexpectedly, the most potent anti-inflammatory effect was registered in the oral administration of the T cells- and B1 cells-derived sEVs. However, no further data is detailed that could explain these findings [49]. When the murine model of DTH was previously depleted of macrophages administrating clodronate liposomes, the anti-inflammatory properties of the T cells-derived sEVs were significantly lost, suggesting that the response of orally administrated T cells-derived sEVs was mediated in part by these myeloid cells [48].

This immunological effect mediated by macrophages after sEVs injection is widely described in the literature since the significant clearance of sEVs in systemic circulation occurs through these myeloid cells [17,50,51,52]. Peyer’s patches (PPs) are subepithelial lymphoid follicles present in the intestine greatly enriched in innate and adaptive immune cells [53]. These dome-shaped clusters control antigen presentation and immunological response by several mechanisms, being the most studied the transendocytosis by specialized epithelial cells named microfold cells (M-cells) towards resident macrophages and dendritic cells [54]. The M-cells have also been previously studied for drug delivery employing synthetic nanoparticles as carrier platforms since these cells have reduced intracellular enzymatic activity and thinner mucus layer and glycocalyx in comparison to enterocytes, promoting easier access and intracellular transport [1]. In the field of drug delivery using nanoparticles, several articles study the active targeting of M-cells through surface modifications, adding peptides [55], mannose receptors ligands [56], and lectin ligands [57], but in most cases, the modification do not fully control unspecific interaction since other cellular lineages express the same receptors. In Table 4 we summarized the studies with cells present in the GI tract that potentially contribute to the uptake of sEVs after oral administration.

Notably, Rubio et al. [23] demonstrated a unidirectional transport from the apical face towards the basal face of fluorescent-labeled *B. subtilis*-derived bEVs in polarized epithelial Caco-2 cells in vitro, where a fraction of those bEVs did not fuse with the cellular membranes and where secreted to the other side. From this study, three major questions arise: (1) if this described transport mechanism is conserved to all sources of sEVs (e.g., bovine milk or human derived EVs); (2) if those secreted “intact” sEVs conserve their capacity to regulate the acceptor cell function and transcriptional expression; (3) if the observed mechanism also occurs in vivo. Regarding the secretion of sEVs from epithelial cells in vivo, Sakhon et al. [58] reported in transgenic mice that M-cells constitutively release sEVs to the subepithelial space, where the highest co-localization was observed with myeloid cells (CX3CR1+/CD11b+ and CX3CR1+/CD11c+ cells). To date, diverse available data concerning the therapeutic mechanism of action of orally administered sEVs partially supports the concept of epithelial transendocytosis and uptake by the immune cells within PPs. However, this idea requires further experiments to confirm it. 

The previously mentioned relationship between epithelial and immune cells from the intestine cannot fully explain the observed phenomena in inflammatory bowel disease (IBD) and cancer models. Tulkens et al. [22] demonstrated that patients with intestinal barrier dysfunction allow the paracellular translocation of bacterial bEVs (diffusion through gaps between adjacent cells), which in the end increased the production of proinflammatory cytokines. This effect was conserved for patients with different treatments (HIV infection, IBD, and chemotherapy), demonstrating that it is vinculated with intestinal barrier dysfunction rather than a particular pathology [22]. Besides, the results published by Samuel et al. [28] showed increased fluorescent signal in tumors of colorectal cancer murine models after oral gavaging of bovine milk-derived sEVs. This result, added to other articles that mention the biodistribution of orally ingested sEVs, suggests that a fraction of the vesicles enter systemic circulation, reaching other tissues and organs, as mentioned above. In a recent review, Ciéslik et al. [24] listed the therapeutic effects of orally administered sEVs from various sources assessed in different diseases. Among the cited pathologies, several articles demonstrate an effect in organs that do not belong to the GI system, suggesting that the ingested circulatory vesicles maintain therapeutic properties [24]. Nevertheless, in most articles that study the therapeutic effect of orally ingested sEVs, only the outcome is measured and characterized without a detailed explanation of a potential mechanism of action that sustains their results.

## 6. Food Derived Vesicles (FDVs)-Based Nutraceutical Perspectives in Infant and Elderly Health

In the last decade, structures morphologically like extracellular vesicles (EVs), called “food-derived vesicles” or FDVs, have been isolated from different foods (such as honey, pollen, milk, fruits, and vegetables, among other foods). These findings raise the question of whether FDVs contain or overexpress the nutritional compounds or nutraceutical effects of the foods from which they are derived. Several studies have shown that different FDVs have nutraceutical effects. For example, it was observed that the nanovesicles in *Apis mellifera* hypopharyngeal gland secretomal products (honey, royal jelly, and bee pollen) participate in the known antibacterial and pro-regenerative properties of bee-derived products [62]. Furthermore, Chen et al. [63] described that honey-derived nanoparticles possess anti-inflammatory properties by inhibiting the NLRP3 inflammasome, thus preventing liver damage in vivo. As the research mentioned, various other studies report the presence of FDVs and their bioactive compounds in different models of diseases such as cancer, intestinal inflammation, and autoimmune diseases [64]. In global terms, the biological effect reported for FDVs is highly associated with the source of identification. However, comparing articles is challenging since different isolation methods are employed, the doses administrated are different, and several routes of administration are tested [36,64].

Several studies have revealed the effects of sEVs derived from breast milk on the immune function of infants. The analysis of breast milk-derived sEVs showed that the molecules they contain vary depending upon the maternal allergy status [65]. Malnutrition in the elderly population is an important risk factor for sarcopenia, osteoporosis, and other age-related diseases. Protein and other components are key nutrients for the human body and affect bone and muscle mass and quality. Dairy products are rich in these nutrients, which implies that dairy products or their bioactive components, such as sEVs might be ideal for the elderly population. The use of milk sEVs as bioactive ingredients represents a novel avenue to explore in the context of human nutrition, and they might exert significant beneficial effects at multiple levels, including but not limited to intestinal health, bone and muscle metabolism, immunity, modulation of the microbiota, growth and development [66].

Due to its nutritional and immunological benefits, bovine milk-derived sEVs have become an essential fraction of proteomic research. An open online database of bovine milk proteome was established, BoMiProt (http://bomiprot.org accessed on 17 november 2022), with over 3100 proteins from whey, milk fat globule membranes (MFGM), and sEVs [67]. Interestingly, ~70% of the research on bovine milk has focused on whey proteins, followed by MFGM (20%) with a surge of sEVs in the past ten years, counting around 10%. The protein lists across milk fractions were compared to identify common and exclusive proteins among whey, MFGM, and sEVs. Interestingly, more than 1300 proteins were exclusively found in exosomes, while 801 and 294 proteins were identified in whey and MFGM, respectively. In contrast, 131 proteins were common across all the fractions [67].

To complement the content characterization, a recent lipidomic study showed eight major lipid classes found in milk sEVs, in which more than 200 fatty acid variations were identified, demonstrating the high complexity of the lipid composition of such vesicles. Also, high levels of phosphatidylcholine (PC), phosphatidylethanolamine (PE), cholesterol (Chol), and phosphatidylserine (PS) were measured. Interestingly, sEVs exhibited increased levels of phosphatidylinositol (PI) as compared to sEVs isolated from cultured cells [68]. Other cargoes, including long noncoding RNAs (lncRNAs) identified in human breast milk sEVs, are also proposed to be implicated in adult metabolism, infant metabolism, neonatal, and development [69].

Many relevant questions need to be answered to understand the value of sEVs in milk formulas since the content of sEVs, and their cargo is modest to not detectable [70]. Dietary depletion of milk sEVs elicited phenotypes such as increased purine metabolites in human and murine body fluids and tissues [71]. Dietary depletion of these sEVs also caused a variety of phenotypes in mice, including a moderate loss of grip strength, an increase in the severity of symptoms of inflammatory bowel disease, a decrease in the postnatal survival, and changes in bacterial communities in the ceca [72]. These observations raise concerns regarding infant and adult nutrition using milk formulas and the need to fortify them with sEVs supplements. However, the compatibility of adding sEVs with existing compounds is crucial for efficient absorption. A recent clinical study by Mutai et al. showed that fortification of soy formulas with milk sEVs must be done by removing lectins for a viable strategy for delivering bioavailable exosomes and their cargos. Lectins in soy formulas bind glycoprotein on the surfaces of milk sEVs, thereby preventing exosome absorption [72].

As a final reflection, this interkingdom relationship has been present throughout the entire existence of animals by dietary consumption, but little was known before the identification of the FDVs.

## 7. Limitation, Future Direction and Conclusions

While oral administration of sEVs presents various physiological and practical advantages over other routes, there is an existing need to investigate further the safety, stability, pharmacokinetics, and biodistribution attributes before their broad use as drug vehicles or nutritional supplements. Most biodistribution studies rely on DiR-based labeling of the sEVs, while DiR labeling does not affect sEVs morphology [73]; however, molecular dissociation between the dye and the sEVs can occur in an in vivo setting and might lead to misleading pharmacokinetic outcomes. Using foreign RNA sequences that quantitative PCR can assess could lead to more precise information devoid of possible artifacts.

The few quality studies presented in the different tables of this review display remarkable features of sEVs being able to resist the harsh acidic environment of the GI tract and reach the intestine. However, most FDVs studies are almost exclusively based on milk-derived sEVs. For this, these findings must be expanded for other FDVs of interest, membrane components, and surface markers depending on the source of isolation. Also, proper readouts must be determined to assess their capacity to elicit significant effects following oral intake. 

So far, milk-derived sEVs are the most feasible option for a drug delivery application based on their published safety profile, pharmacokinetics, and endogenous stability in the GI tract. Nonetheless, this does not mean, in any case, that these properties are exclusive for this source of sEVs. As we mention in this article, studies evaluating other sources of vesicles in detail could discover an optimal sEVs-based nanocarrier for oral drug delivery. 

It is interesting to mention that oral administration is getting closer to clinical-stage application. So far, we have found only one clinical trial seeking to characterize the ability of plant-derived exosomes in the delivery of curcumin to colon tissue in the context of colon cancer via oral intake (ClinicalTrials.gov Identifier: NCT01294072). This study is ongoing, and no data or results are available yet.

Several preclinical and clinical studies have failed to show a positive association between milk intake and serum miRNA levels. However, the role of miRNA cargo, the extent to which they are exported via the sEVs route, and whether they contribute to cell-cell communication are still controversial. Albanese et al. found that sEVs did not fuse detectably with cellular membranes to deliver their cargo. They engineered sEVs to be fusogenic and documented their capacity to deliver functional messenger RNAs. Engineered fusogenic sEVs, however, did not detectably alter the functionality of cells exposed to miRNA-carrying sEVs. These results suggest that sEVs-borne miRNAs do not act as effectors of cell-to-cell communication, suggesting that the delivery of different RNA species through the sEVs might be an extremely inefficient process [74].

The interaction of xeno-EVs (plant or bovine to human cells) adds another layer of complication to deciphering the fusion and cargo delivery processes, identifying the glycan features and other pathways on the EVs surface responsible for the interaction between plant/animal EVs and human cells. In conclusion, oral administration presents a reliable delivery route for EVs. The pharmacokinetics and activity of these bioactive compounds will depend on their cellular source, cargo content, and interaction with human cells.

## Figures and Tables

**Figure 1 pharmaceutics-15-00716-f001:**
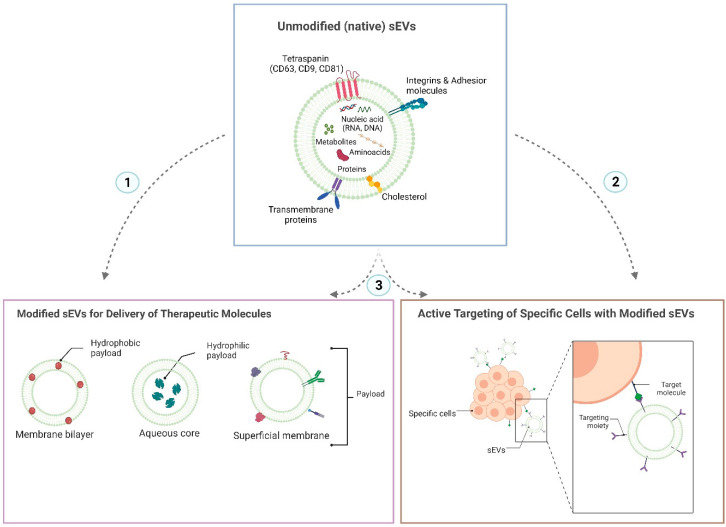
Diagram of the native structure of small extracellular vesicles and the functionalization strategies that can be performed on them to provide them with specific therapeutic properties. Small extracellular vesicles (sEVs) have a structure formed by a membrane composed of a lipid bilayer. Different proteins are expressed in it, which may be common to the vast majority of sEVs (such as tetraspanins for sEVs derived from eukaryotic cells) or specific proteins according to the origin of their parental cell. The core of sEVs is composed of nucleic acids, lipids, proteins, and metabolites. One of the characteristics of sEVs that make them good nanocarriers is that they can be easily modified to endow them with specific therapeutic properties. For example, to acquire a certain therapeutic efficacy, sEVs can be engineered to carry a specific therapeutic payload: drugs, proteins, or different types of nucleic acids (siRNA, miRNA, shRNA). Depending on the molecule’s type and therapeutic function to be triggered, the payload can be incorporated into or anchored to the surface of the sEVs membrane. It can also be loaded into the sEVs core (1). To provide them with a better safety profile, sEVs can be functionalized to target a specific cell or tissue by incorporating a targeting moiety into their surface membrane (2). This strategy reduces off-target interactions while improving the bioavailability of the therapeutic molecule at the site of interest. Both the strategy of therapeutic loading molecules and the strategy of targeting sEVs to a specific tissue can be performed together in sEVs (3), providing the nanovesicles with better efficacy and safety profiles at the same time (created with http://www.biorender.com (accessed on 16 november 2022)).

**Figure 2 pharmaceutics-15-00716-f002:**
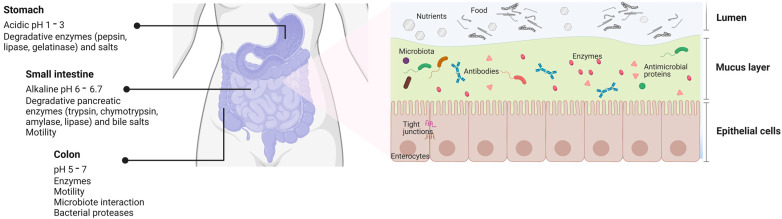
Scheme of the gastrointestinal tract and the physiological factors that influence the absorption of therapeutic molecules. Several physiological barriers in the gastrointestinal (GI) tract challenge drug administration by the oral route. In the GI environment, the presence of factors such as pH, degradative enzymes and salts, motility and interaction with the microbiota can alter the solubility and stability of drugs, which finally affect their permeability across the mucosal barriers. This figure is based on a schematic drawing and does not fully represent the accurate structural reality of the intestine (created with http://www.biorender.com (accessed on 21 november 2022)).

**Figure 3 pharmaceutics-15-00716-f003:**
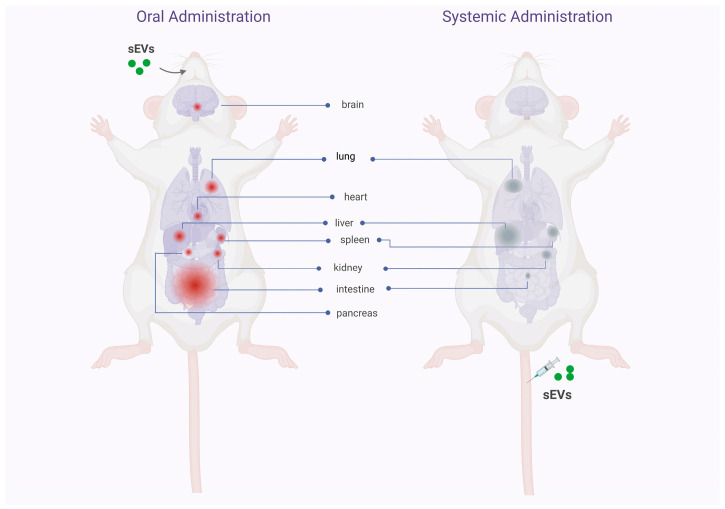
Comparative diagram of the biodistribution pattern of sEV administered orally and intravenously. The illustration shows the pattern of biodistribution of sEVs in different mice tissues after oral or intravenous administration. In the body on the left, the tissues and organs where the sEV would accumulate after intestinal absorption are identified in red. The considerable accumulation of sEVs in the intestine and, to a lesser extent, in the rest of the body’s organs stands out. In the body on the right, the organs where sEVs would accumulate after intravenous administration are identified in gray. A considerable accumulation of sEVs is observed in the organs associated with the mononuclear phagocytic system (liver, spleen, lung), with little reach to other body organs. These data suggest that the biodistribution pattern is defined by the route of administration of the sEVs, a dependency that can be used strategically to reach a specific organ in patients (created with http://www.biorender.com (accessed on 2 february 2023)).

**Figure 4 pharmaceutics-15-00716-f004:**
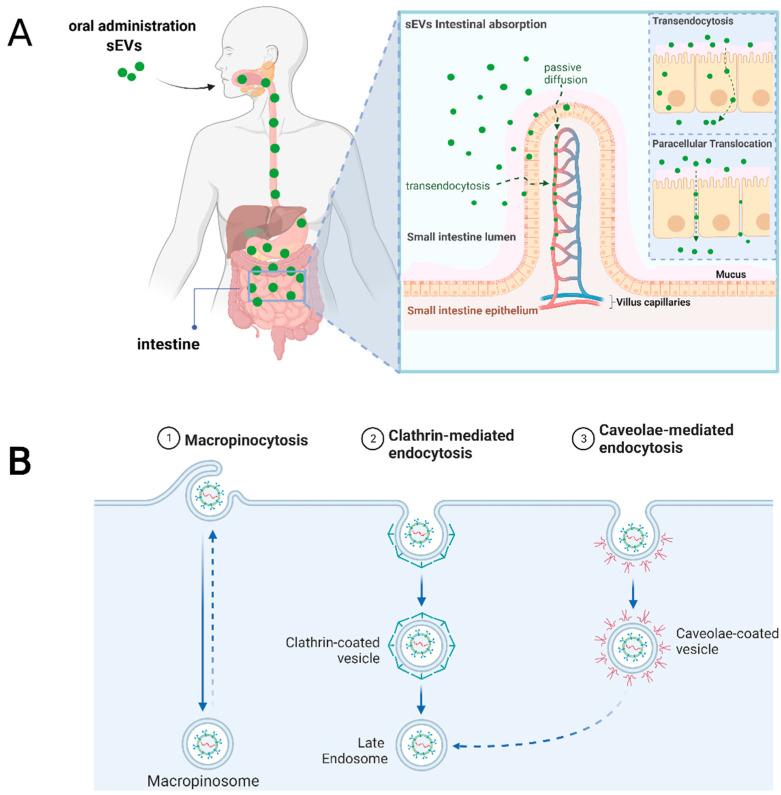
Scheme of the cellular mechanisms of absorption in the gastrointestinal tract of orally administered sEVs. (**A**) After oral intake, sEVs can resist the hostile degradative environment of the gastrointestinal tract and reach the intestine. At this site, the absorption of sEVs or their transport into the systemic circulation would occur, overcoming the cell permeability barrier. This figure was constructed based on preclinical in vivo/in vitro revised data and is a theorical extrapolation of what is expected in human. (**B**) In vitro studies demonstrate that multiple endocytic pathways are involved, including caveola- and clathrin-mediated pathways and macropinocytosis. Additionally, it has been described that sEVs could cross the intestinal epithelium through the intercellular spaces between epithelial cells, passively transported from the intestinal lumen to the circulation in a mechanism called paracellular translocation (created with http://www.biorender.com (accessed on 24 November 2022)).

**Table 1 pharmaceutics-15-00716-t001:** Comparative summary of the biodistribution of orally administered sEVs.

Type of sEV	Cell Source	Labeling Method	Dose	Mouse Strain	Time of Detection and Tissue Distribution	Reference
sEVs	Bovine milk	DiR	0.5 mg prot/mouse	C57BL/6 mice with DSS-induced ulcerative colitis	1 h Small intestine 6 h Colon	[27]
Exosomes	Bovine milk	DiR	1 × 10^12^ part/gr of mouse	Balb/c mice	24 h Intestine, lung, and liver	[30]
Exosomes	Bovine milk	DiR	40 mg prot/kg of mouse	Balb/c mice	30 min Blood 6 h Liver, spleen, kidney, heart, and lung	[29]
iRGD-Exosomes	Bovine milk	DiR	40 mg prot/kg of mouse	Tumor-bearing Balb/c mice	4 h Tumor, liver, spleen, kidney, lung, and heart	[29]
Exosomes	Bovine milk	DiR	60 mg prot/kg of mouse	Athymic nude mice	4 d Liver, lung, kidney, pancreas, spleen, ovaries, colon, and brain	[31]
sEVs	Bovine milk	DiR	25 mg prot/kg of mouse	Balb/c mice	2 and 6 h Intestine 24 h GI tract, liver, spleen, lungs, kidney, and heart	[28]
sEVs	Bovine milk	DiR	25 mg prot/kg of mouse/day × 38 days	Balb/c-Fox1nuAusb mice	24 h Tumor tissue	[28]
sEVs	Yeast	DiR	25 mg prot/kg of mouse	Not indicated	24 h GI tract, liver, spleen, kidney, lungs, and heart	[28]
sEVs	Beer	DiR	Not indicated	Not indicated	Not bioavailable in the mice	[28]
Exosomes-like	Grape	DiR, PKH26	1 mg/mouse	C57BL/6 mice	6 h Intestine	[34]
Exosomes-like	Acerola	PKH26	3 × 10^9^ particles/mouse	C57BL/6 mice	1 h Intestine, liver, and bladder, weak signal in brain	[33]
Exosomes-like	Ginger	DiR	0.3 mg/mouse	C57BL/6 mice	12 h Colon (in non-starved mice); 12 h Stomach and small intestine (in starved mice)	[32]
Exosomes-like	Garlic	DiR PKH26	1 × 10^10^ particles/mouse	C57BL/6 mice	24 h Brain, liver, small intestine, and large intestine	[35]
Exosomes-like	Tea leaves	DiR	3 mg/kg of mouse	Balb/c mice	6 h Small intestine	[36]
Exosomes-like	Mulberry bark	DiR	10 × 10^10^ particles/mouse	C57BL/6 mice	3 h Small intestine, colon, cecum; small fraction was observed in spleen, liver, lung, kidney, heart, and blood	[37]

Abbreviations: sEVs, small extracellular vesicles; DIR, DiIC18(7); 1,1′-dioctadecyl-3,3,3′,3′-tetramethylindotricarbocyanine iodide; part, particles; prot, protein; gr, grams; kg, kilograms; DSS, dextran sodium sulfate; min, minutes; h, hours; d, days; GI tract, gastrointestinal tract.

**Table 2 pharmaceutics-15-00716-t002:** Summary of the main findings related to toxicity studies after oral intake of sEVs.

sEVs Type and Cell Source	Dose	Murine Strain	Time of Detection	Toxicity Profile	Reference
Cow’s milk-derived exosomes	25 mg/kg (single administration) 25 mg/kg daily × 15 d	Sprague Dawley rats	6 h 15 d	No changes in clinical signs, body weight, or dietary intake in animals. Biochemical (liver and kidney function) and hematological parameters remained unchanged except for triglycerides. No changes in cytokine profile (IL-1α, IL-1β, IL-2, IL-4, IL-5, IL-6, IL-10, IL-12, IL-13, GM-CSF, IFN-γ and TNF-α), except for the anti-inflammatory cytokine GM-CSF.	[31]
Cow’s milk-derived sEVs	2 mg/kg × 7 d	IRC mice	7 d	No changes in body weight in animals. Biochemical (liver function) and hematological parameters remained unchanged. Histopathology examination (H&E staining) of the heart, liver, spleen, lung, kidney, and small intestine exhibited no pathological changes.	[25]

Abbreviations: sEVs, small extracellular vesicles; mg, miligrams; kg, kilograms; h, hours; d, days.

**Table 3 pharmaceutics-15-00716-t003:** Summary of sEVs reported attributes for oral administration.

	sEV Source	sEV Attribute/Modification/Treatment	Loaded Molecule	Biological Effect	Type of Study	References
**sEVs protection in transit trough digestive tract**	Bovine milk and colorectal cancer cells	Calcium chloride addition	N/A	Enhanced EV stability after acidification (pH = 2) and boiling (105 °C)	In vitro	[28]
	Human cardiosphere-derived stromal cells	Casein addition	N/A	Enhanced uptake and disease-modifying bioactivity	In vivo	[43]
**sEVs uptake by gut cells**	Grape juice	Phosphatidic acids, Phosphatidylethanolamines	N/A	Dextrane sulfate sodium-induced colitis protection via induction of intestinal stem cells	In vivo	[34]
**sEVs targeting beyond gut mucosa**	Bovine milk	Non modified	N/A	Tumor growth reduction and accelerated metastasis, xenograft	In vivo	[28]
	Bovine milk	Folic acid functionalization	Withaferin A Anthocyanidins Curcumin Paclitaxel Docetxel	Tumor targeting, xenograft	In vivo	[31]
	Human umbilical cord	Non modified	N/A	Antioxidant and anti-apoptotic and rescue from liver failure	In vivo	[44]
	Human MSC544 cell line	Non modified	Taxol	Tumor reducing capabilities	In vivo	[45]
	Mouse suppresor T cells	Antibodies free light chains coating	miRNA-150	Immune tolerance	In vivo	[46]

Abbreviations: sEVs, small extracellular vesicles; N/A, not applicable.

**Table 4 pharmaceutics-15-00716-t004:** Summary of cellular and molecular mediators of sEVs uptake in oral administration.

Cell Lineage	Target Cell Source	sEV Source	Findings	Type of Study	References
Microfold cells (M cells)	Several sources	N/A (Synthetic nanoparticles)	M cells possess reduced intracellular enzymatic activity, thinner mucus layer and glycocalyx, promoting easier access and intracellular transport.	In vitro In vivo	[1]
Macrophages	Ag-presenting macrophages	Ts cell-derived	Macrophage clodronate depletion abolishes anti-inflammatory effect of Ts derived sEVs observed in DTH model.	In vivo	[48]
Dendritic cells	Transgenic reporter mice	M cell-derived vesicles	M cell-derived vesicles are taken up by dendritic cells.	In vivo	[58]
Enterocytes	Rat intestinal epithelial cells (IEC-6)	Grapefruit juice	Plant EV’s miRNAs are taken up by rat intestinal enterocytes.	In vitro	[59]
Colonocytes	Human colonocyte cell line (DLD-1) Mouse intestinal epithelial cell line (CMT-93)	Colonic luminal fluid aspirates	sEVs mRNA was present within cells, showing take up.	In vitro	[60]
Enterocytes	Porcine intestinal cells (IPEC-J2)	Porcine milk	sEVs promoted enterocytes proliferation in vitro. increased villus height, crypt depth and ratio of villus length to crypt depth of intestinal tissues was observed in vivo.	In vitro In vivo	[61]
Enterocytes	Rat intestinal epithelial cells (IEC-6) Human colon carcinoma (Caco-2)	Bovine milk	sEVs uptake decreased when incubated at low temperature (4 °C), after proteinase K treatment, using endocytosis inhibitors or carbohydrate competitors.	In vitro	[38]

Abbreviations: Ag, antigen; DTH, delayed-type hypersensitivity; Ts, suppressor T cell; N/A, not applicable.

## Data Availability

The data that support the findings of this study are available from the corresponding author upon reasonable request.

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
