# Peer review of "Oral Administration as a Potential Alternative for the Delivery of Small Extracellular Vesicles"

_pharmaceutics, 2023, doi:10.3390/pharmaceutics15030716_

Round 1
Reviewer 1 Report
The authors present this review about the potential role in the delivery of small extracellular vesicles. In my opinion, this review is well write and all points considered are well described.
Reviewer 2 Report
The authors submitted a very nice review to Pharmaceutics. The paper is of high quality, but the authors should resolve several issues before the review can be accepted.
Major issues
- According to the Tables 1 and 3. Were any experiments on the biodistribution of oral delivery conducted on the EVs or exosomes isolated from sources other than Bovine milk? The results of Yeast and Beer exosomes are shown in Table 1. The analysis of EV modifications is presented in Table 3. But were the vesicles from cell culture liquids and/or other sources analyzed?
- The other Table worth to be added to the review may content the molecules which oral delivery by EVs were analyzed.
- Figures 3 and 4b show the human body, but the experiments were conducted on mice. Such images can mislead the reader. Table 1 contains the only experiments conducted on mice.
- I would like to invite the authors to add comparative analysis and express their opinion on which source of vesicles is optimal for oral delivery.
Minor issues
- Quality of the Figures is very poor
- The sentence on line 131 should start a new paragraph
- Microvilli in Fig 2 and Fig 4a are shown to be distributed at a considerable distance from each other, while the literature data, TEM, and SEM show them touching each other.
Reviewer 3 Report
The review article "Oral administration as a potential alternative for the delivery of small extracellular vesicles", is an interesting topic and at the same time provides data from the specialist literature regarding small extracellular vesicles (sEVs), which can be used as delivery systems the release of new active molecules.
Reviewer 4 Report
This review by Donoso-Meneses describes the future possibility of using small extracellular vesicles (sEVs) for delivery of therapeutic molecules by the oral route. The authors talk about the challenges of oral delivery, biodistribution, stability and safety, mediators of small sEVs and potential sources of sEVs for therapeutic use. The review is timely and has covered a good part of the literature. Some details about the oral efficacy and safety of sEVs are missing and addressing them is important.
1. While the authors discuss the pharmacokinetics, there is still discussion about how the proteins present on the outer membranes of sEVs survive the digestive enzymes as soon as they enter the duodenum.
2. With regards to source of sEVs for therapeutic use, what steps can be taken to ensure that the isolated sEVs are largely devoid of nucleic acids such as non-coding RNAs and proteins that may cause cellular transformation in host cells.
3. There are significant English language issues that need to be addressed with the help of an English language expert.
Round 2
Reviewer 2 Report
Dear Authors, dear Editor!
Аfter reading the authors' replies, I still have doubts that the article can be published in the current version with the current figures.
1) According to the Fig. 3 and Fig. 4 - I'm not sure that data obtained in mice experiments can be "extrapolated to humans" in such an illustrative form, this misleads the reader.
2) Ths Biorender is not the way to obtain "professional quality figures" and the Fig. 2 is such an example - the structure of microvilli IS WRONG, and it misleads the reader, since the exosomes and other sEV obviously cannot reach the membranes due to the glycocalyx.
3) Unfortunately, changes in the manuscript highlighted in red is the worst option to help the colorblind reviewer "easily identify" edits. I recommend using highlighting of text background.
Sincerely
